# Layer-by-Layer Assembly of Polystyrene/Ag for a Highly Reproducible SERS Substrate and Its Use for the Detection of Food Contaminants

**DOI:** 10.3390/polym13193270

**Published:** 2021-09-25

**Authors:** Sihan Zhang, Zhihua Xu, Jiaqi Guo, Haiying Wang, Yibo Ma, Xianming Kong, Hongtao Fan, Qian Yu

**Affiliations:** 1School of Petrochemical Engineering, Liaoning Petrochemical University, Fushun 113001, China; h321065125@163.com (S.Z.); kxm.kxm@163.com (Z.X.); httyf_77@163.com (H.F.); 2Jiangsu Co-Innovation Center for Efficient Processing and Utilization of Forest Resources and Joint International Research Lab of Lignocellulosic Functional Materials, Nanjing Forestry University, Nanjing 210037, China; jiaqi.guo@njfu.edu.cn; 3School of Environmental Science, Nanjing Xiaozhuang University, Nanjing 210037, China; wanghaiying@nju.edu.cn; 4Department of Bioproducts and Biosystems, School of Chemical Engineering, Aalto University, FI-00076 Aalto, Finland; yibo.ma@aalto.fi

**Keywords:** polystyrene, self-assembly, Ag nanoparticle, layer by layer, SERS, reproducibility

## Abstract

Polystyrene (PS) spheres were prepared through an emulsifier-free emulsion polymerization method, in which the reaction time, ionic strength, concentrations of copolymer were studied in detail. The resulting PS microspheres and Ag nanoparticles were used to construct a surface enhanced Raman scattering (SERS) substrate by a layer-by-layer assembly method. A relatively uniform distribution of PS/Ag in the films was obtained, and the multilayer substrate presented excellent SERS reproducibility and a tunable enhancement effect. The SERS substrate was used for detecting harmful pesticides (malachite green and dimetridazole) in food samples, with a limit of detection as low as 3.5 ppb. The obtained plasmonic composite has a promising future in the field of SERS sensing.

## 1. Introduction

Polystyrene (PS) materials have attracted considerable attention due to their multiple excellent properties such as strong adsorption capacity, oxygen permeability and the possibility of surface functionalization. PS microspheres and their composites have been widely used in biochemistry, analytical chemistry, chromatographic separation, catalysis and other fields [1,2,3]. Liang’s group has fabricated a new type of PS microspheres containing rhodamine and a platinum porphyrin, which presented good oxygen sensitivity and light stability during in situ monitoring of cellular oxygen respiration and analysis of dissolved oxygen concentrations in beverages [4]. Sharma and Kaur decorated polystyrene resin with Au NPs, The PS@Au composite shown good catalytic performance in the 1,4-dioxane oxidation process [5]. 

The preparation of the PS colloid is a key factor for the construction of PS composites. Numerous methods have been proposed for preparing PS microspheres, such as suspension polymerization, emulsion polymerization, emulsifier-free emulsion polymerization (EFEP), seed polymerization and dispersion polymerization [6,7,8]. The key benefits of EFEP are that the product has excellent monodispersity, high purity, uniform particle size and the process is simple. Therefore, this method has become the common method for preparing polystyrene and many other functional polymer materials. For example, Abdollahi’s group prepared PS latex particles modified with different functional groups by EFEP, and observed that PS latex particles with different surface functional groups have different morphologies through SEM and transmission electron microscopy (TEM) [9]. Tumnantong et.al. fabricated hybrid nanoparticles of polystyrene and silica with a core-shell structure by the EFEP method, in which sodium styrene sulfonate was chosen as initiator. These polyisoprene-silica nanoparticles were used as filler in rubber materials and exhibited excellent mechanical features, oil resistance and thermal resistance [10].

SERS spectroscopy is an advanced analytical method, that could identify target molecules at trace concentrations from fingerprint information. The SERS method can obtain spectra of analytes at a single molecule level [11,12,13]. The high sensitivity of SERS is mainly due to the localized surface plasmon resonance (LSPR) of the SERS substrate. Various plasmonic composites were developed and used as SERS substrates [14,15,16,17,18,19,20,21]. The noble metal nanoparticle-functionalized PS microspheres have attracted considerable interest as SERS substrates due to their uniform micrometer size supporting well defined plasmonic nanoparticles on the surface. The PS spheres function as stable supports for producing the ‘hot spot’ in SERS measurement. In order to obtain the excellent properties of the SERS substrate base in a PS composite, it is important to improve the distribution of plasmonic NPs and PS spheres. Considerable research effort has been invested in the deposition of plasmonic NPs onto PS spherea. Cai et al. has deposited dense gold NPs on PS microspheres and used the resulting composite as a SERS substrate. PS microspheres with a size of 1.5 μm were synthesized through dispersion polymerization and sulfonated with concentrated sulfuric acid [22]. Li and co-workers prepared a PS/Ag nanocomposite and used it as a dynamic-SERS platform for sensing organophosphorus pesticides, in which the limit of detection for paraoxon could reach 96 nM [23]. The PS/Ag nanocomposite was fabricated through a three-step procedure. It is well-known that uniformity is important in SERS measurement, and that it is challenging to reproduce the absolute SERS intensity of analytes with a colloidal PS composite as it is difficult to control the distribution of plasmonic PS composites after being dropped for Raman signal collection. SERS substrates with 2-dimensional (2D) structure are promising to provide high enhancement and good reproducibility. The layer-by-layer (LBL) assembly technique is a universal, convenient and efficient method for fabricating 2D nanostructures with controllable morphology and properties [24,25,26]. The films fabricated by LBL technology mainly depend on the electrostatic adsorption between the opposite charges, in which the dimensional precision could down to the submicron level [27]. Kahraman et al. deposited plasmonic NPs on bacterial cells by a layer-by-layer method, in which the Raman information of single bacterial cella was successfully obtained [28].

DMZ is a kind of veterinary drug belong to the 5-nitroimidazole derivatives group that are commonly used for healing protozoal infections in poultry and swine [29]. Residue of DMZ in meat or the environment could bring harmful effects to public health as it is carcinogenic and mutagenic. Therefore, the DMZ residues are strictly monitored in food samples, and the detection of DMZ is important for protecting public health. Pork is a good source of protein and is a favorite food in many countries. DMZ residues in pork would therefore bring harmful health problems to a large number of consumers.

In this study, PS spheres were firstly synthesized through an EFEP method, in which the size of the PS spheres is controllable by adjusting the parameters in the synthesis process. Polydimethyldiallyl ammonium chloride (PDDA), PS spheres and Ag NPs were used as matrix for the facile assembly of a LBL SERS substrate. A multilayer PS/Ag nanostructure with excellent SERS sensitivity and high uniformity was obtained through the simple and controllable assembly process. The sensitivity and signal repeatability of LBL PS/Ag is better than that of colloidal PS/Ag in SERS measurements. The LBL PS/Ag was further used in detecting metronidazole in a real food sample (meat) by SERS.

## 2. Experimental Section

### 2.1. Chemicals and Reagents

Sodium styrene sulfonate (C_8_H_7_SO_3_Na) was obtained from Admas-Beta Co., Ltd. (Shanghai, China). AgNO_3_, potassium bicarbonate (KHCO_3_) and potassium persulfate (K_2_S_2_O_8_) were supplied by Innochem Sci. & Tech. Co., Ltd. (Beijing, China). Sodium citrate was purchased from Macklin Reagents (Shanghai, China). Styrene was obtained from Damao Chemical Reagent (Tanjin, China). PDDA and 4-mercaptobenzoic acid (4-MBA) were obtained from Aladdin (Shanghai, China). Ultrapure water used in all experiments. All reagents were used as received without further purification.

### 2.2. Apparatus

Scanning electron microscope (SEM) images were acquired with a SU8010 field emission scanning electron microscope (Hitachi, Tokyo, Japan). Fourier transform infrared (FTIR) spectra were collected on a Nicolet 6700 spectrometer (PerkinElmer, Waltham, MA, USA) equipped with a DTGS detector and using the KBr method. Typically 32 scans were collected to obtain a satisfactory signal-to-noise ratio with a resolution of 4 cm^−1^. The Raman signals were collected by a portable spectrometer (BWS465 iRman; B&W Tek, Newark, DE, USA). A 785 nm laser was used as excitation light, and the spectra acquisition time was 2 s with 5 cm^−1^ resolutions. After SERS measurement the substrates were burned in 500 °C to exclude the nanoplastics. UV-vis absorption spectra were measured on a UV2400 UV–Vis spectrophotometer (Sunny Hengping Instrument, Shanghai, China) in quartz cells with 1 cm optical path.

### 2.3. Synthesis of Ag Colloid and PS Spheres

The synthesis of Ag colloid was performed according to a previous report [30]. The specific operation process was as follows: first, all glassware used in this experiment was cleaned with aqua regia (HNO_3_/HCl, 1:3, *v*/*v*) and rinsed thoroughly with distilled water. Next 200 mL of aqueous solution of AgNO_3_ were heated to reflux, and sodium citrate solution (4 mL, 1%, *w*/*v*) was quickly added into the boiling system. The reflux was continued for another 1 h, and then the mixture was cooled to room temperature.

The PS spheres were prepared through the typical EFEP method [31]. Initially, KHCO_3_ and C_8_H_7_SO_3_Na were added to 100 mL of water. Then, the solution was transferred into a flask, followed by the addition of styrene (13 mL). A 25 mL aqueous solution of K_2_S_2_O_8_ (0.037 M) was preheated to 72 °C and added dropwise into the reaction system over half an hour. After a certain reaction time, the system was cooled to room temperature, and the resulting PS spheres were centrifuged and kept in the refrigerator for further use. 

### 2.4. Fabrication of Colloidal PS/Ag Nanocomposite

The fabrication procedure of colloidal PS/Ag nanocomposite was as follows: The prepared PS spheres were dried in an oven. Then, 50 mg of PS spheres were dispersed in 4 mL of aqueous solution of PDDA (1%) and kept shaking for 1 h. The PS spheres were centrifuged and washed thoroughly with water, and a certain amount of PDDA-treated PS spheres (10 mg/mL) were added into 1 mL of silver colloid. The self-assembly procedure was performed at room temperature for 3 h. After centrifugation, colloidal PS/Ag composite was obtained. 

### 2.5. Layer-by-Layer Assembly of Ag NPs and PS Spheres

Glass slides were used as solid support and cleaned with ethanol and water before the assembly process. The glass slides were soaked in an aqueous solution of PDDA (1%) for half an hour and washed with water. After that, the substrate was immersed in 1 mL of PS colloid for 4 h. This is the first layer of PS. The substrate was subsequently immersed in aqueous solution of PDDA (1%) for half an hour and rinsed thoroughly with water. Then, the substrate was immersed in Ag colloid (1 mL) for 4 h, this is the second layer 2D structure of the PS/Ag composite. A multilayer PS/Ag was fabricated by alternative repetition of the above steps.

## 3. Results and Discussion

### 3.1. The Effect of Reaction Conditions on the Diameter of PS Spheres

Initially, 0.075 g of KHCO_3_ and 0.013 g of C_8_H_7_SO_3_Na were added to distilled water (100 mL). Then, the mixture was transferred to a 4-neck flask, followed by the addition of styrene (13 mL). A 25 mL aqueous solution of K_2_S_2_O_8_ (0.037 M) was preheated to 72 °C and added into the reaction system over 30 min. The size distributions of PS spheres with various reaction times are shown in Appendix A. The average diameters of PS spheres corresponding to 6, 8 and 10 h reaction time were 0.77, 0.76 and 0.77 μm, respectively. The extension of polymerization time beyond 6 h won’t change the size of PS sphere obviously. The result indicated that the styrene monomer was polymerizing and increasing the size of the PS spheres during the early stage of the reaction [32], and 8 h was chosen as the best reaction time for further investigation. 

Figure 1 shows SEM images of PS spheres prepared with different dosages of KHCO_3_ and C_8_H_7_SO_3_Na, respectively. Concentrations of KHCO_3_ of 4.5, 7.5, 10 and 17 mM corresponded to 450, 750, 790 and 830 nm diameter spheres, respectively. The particle size increases with the increase of ionic strength. The particle size of PS spheres also depended on the concentration of C_8_H_7_SO_3_Na. The increment in the concentration of C_8_H_7_SO_3_Na results in a smaller diameter of PS spheres. The oligomer free radicals and sulfonic acid groups were increased as the concentration of C_8_H_7_SO_3_Na increased, providing more active centers during the polymerization process and producing PS spheres with a smaller diameter [33].

FTIR spectroscopy was used to characterize the surface groups of the PS spheres. As shown in Figure 2, the narrow bands at 697 cm^−1^ and 757 cm^−1^ are assigned to out-of-plane bending vibration of the C–H bond of PS spheres. The characteristic peaks at 1450 cm^−1^, 1495 cm^−1^ and 1601 cm^−1^ correspond to the C=C stretching vibration of benzene ring, the peak at 3027 cm^−1^ is attributed to the vibration stretching of C–H in aromatic. The peaks at 2852 cm^−1^ and 2922 cm^−1^ are assigned to the symmetric and antisymmetric stretching vibrations of the methylene group [34]. After grafting the PDDA on the surface of PS spheres, there is nearly no difference of the intensity and position of the FTIR bands. The result indicated that the modification of PDDA on surface of the PS sphere would not change the surface groups of the PS.

### 3.2. Characterization of PS/Ag Composite 

The decoration of Ag NPs on PS sphere was achieved via a self-assembly process. The citrate-capped Ag NPs could be easily adsorbed on the surface of PDDA-modified PS spheres through electrostatic interactions. The coverage of Ag NPs on the PS spheres could be controlled by adjusting the amount of PS colloid added to the Ag colloid. Appendix A exhibits the SEM image of Ag NPs-coated PS spheres fabricated with different amounts of PS. When the volume of PS colloid was 400 μL, only a few Ag NPs were observed on the surface of the PS spheres, as shown in Appendix A. As the amount of PS colloid decreased, the amounts of Ag NPs on the PS spheres increased as shown in Appendix A. The SERS performance of the prepared colloidal PS/Ag was investigated using MBA (100 ppm) as Raman reporter. As shown in Appendix A, an increasing Raman intensity of MBA is obtained with decreasing amount of PS colloid, which is due to the fact a high density of Ag NPs on PS could provide more ‘hot spots’ for SERS measurements.

### 3.3. The Multilayer PS/Ag Substrate

The surface morphologies of the 2D PS/Ag substrate were firstly determined through SEM. Representative SEM images are shown in Figure 3. The SEM image of four layer PS/Ag is presented in Figure 3a, in which the PS spheres with a diameter nearly at 800 nm were distributed on the substrate, and the low coverage of Ag NPs on the 2D substrate is visible. When the number of layers was increased to six (Figure 3b), more Ag NPs were decorated on the surface of PS, and the density of particles on the substrate was obviously increased. When the number of assembly cycles was increased to 10, more PS and Ag NPs were patterned on the substrate as shown in Figure 3c. The PS and Ag NPs are thickly and densely distributed on the surface, and the surface coverage of Ag NPs is greatly increased. When the number of layers continues to be increased to 15, the corresponding SEM image is shown in Figure 3d, where the density of PS was increased compared with 10 layers. However, the coverage of Ag NPs was decreased, which is due to the fact the amount of Ag NPs left in the fixed volume of Ag colloid was decreased. 

MBA is commonly used as Raman reporter for evaluating the enhancement performance of the SERS substrate; it can easily crosslink onto the surface of metallic material substrated as chemical bonds are formed between its thiol group and metals [35]. Here, MBA was used to evaluate the SERS enhancement of the 2D PS/Ag substrates with multilayer structure. Five μL of MBA solution was dropped onto the surface of a LBL PS/Ag substrate. The SERS spectra of MBA (100 ppm) from a 2D multilayer PS/Ag substrate are shown in Figure 4. There are two prominent Raman peaks observed at 1073 and 1583 cm^−1^. The peak at 1073 cm^−1^ is due to the in-plane ring breathing vibration of the aromatic ring; the peak at 1583 cm^−1^ is assigned to the totally symmetric vibration of the C–C group of the ring [36,37]. The intensity of the SERS signal of MBA increased in going from four layers to 10 layers and reached a maximum at 10 layers. The increment of the intensity of SERS spectra is due to the dense Ag NPs on the substrate. The junctions between inter-particle of Ag NPs provide ‘hot spots’ in SERS sensing. The SERS intensity dropped when the PS/Ag was increased to 15 layers. The reason is that the amount of Ag NPs left in the Ag colloid used for assembly is lower, and the density of Ag NPs on the PS was decreased, which coincides with the SEM image results, indicating that the substrate with 10 layers of PS/Ag achieves the optimal SERS enhancement.

The uniformity of SERS signals collected for a substrate is critical for any practical application of SERS. The reproducibility of the colloidal PS/Ag and 2D PS/Ag were evaluated by measuring the SERS signals of MBA (100 ppm) from different positions on the SERS substrate. Ten μL of colloidal PS/Ag was dropped onto a glass substrate and dried under air, then 5 μL of MBA solution (100 ppm) was dropped onto the colloidal PS/Ag and used for Raman measurement. The SERS spectra of MBA from 10 random spots on the colloidal PS/Ag and 2D PS/Ag are presented in Figure 5. The SERS spectra of MBA on the colloidal PS/Ag are shown in Figure 5a, the relative standard deviation (RSD) of the Raman peak intensity was employed to evaluate the reproducibility of SERS spectra on the substrate. Figure 5b quantitatively shows the intensity variation of the Raman peaks at 1073 cm^−1^, in which the RSD value is 13.7%. The SERS spectra of MBA from 10 random spots from 2D PS/Ag assembled by LBL are shown in Figure 5c. The average intensity is two times higher than that from the colloidal PS/Ag. Another merit of these 2D PS/Ag substrates is the relatively uniform distribution of Ag NPs in the multilayers. The RSD value calculated from the ten SERS peaks at 1073 cm^−1^ from 2D PS/Ag is 3.7% (Figure 5d). These results indicate that the LBL 2D PS/Ag substrate could provide excellent SERS sensitivity and reproducibility [38].

MG is a kind of triphenylmethane dye, that is usually employed as a dye in the textile or paper industry. MG is also an effective fungicide for healing fish diseases. However, abusive use of MG could bring health problems due to its genotoxic and carcinogenic properties, therefore, the use of MG as a fungicide is prohibited in the European Union, the United States, China, and Japan [39]. The 2D PS/Ag was used as SERS substrate for MG sensing, whereby 5 μL of aqueous solution of MG at different concentrations were dropped onto LBL PS/Ag substrate. The SERS spectra of MG measured from the LBL PS/Ag substrate is shown in Figure 6a. The characteristic Raman peaks of MG were observed at 439, 796, 913, 1169, 1392 and 1616 cm^−1^ [40]. The intensity of the characteristic Raman peaks of MG decreased as the concentration decreased. A linear regression curve was built between the concentration of MG and SERS signal intensity at 1616 cm^−1^ as shown in Figure 6b. The obvious characteristic peaks of MG could still be seen as the concentration went down to 10^−7^ M.

The LBL PS/Ag was used as SERS substrate for DMZ detection, where different concentrations of DMZ were dropped onto LBL PS/Ag substrate. The SERS spectra of DMZ are shown in Figure 7a. Obvious Raman peaks at 796, 937, 1298 and 1390 cm^−1^ were observed, which are assigned to the characteristic peaks of DMZ [41]. A linear regression curve was built between the concentration of DMZ and the SERS signal intensity at 1298 cm^−1^ as shown in Figure 7b, the equation between concentration of DMZ from solution and Raman intensity is y = 4074.8 + 1489 × logX. The limit of detection was defined as a SNR ratio at three, and that is nearly at 3.5 ppb. That is lower than the maximum residue limit of DMZ in edible meat (5 ppb) [42].

The LBL PS/Ag was used as SERS substrate for rapidly detecting DMZ in pork. Solutions of DMZ at different concentrations were sprayed onto the surface of pork. Then 5 μL of juice from the pork was spotted onto the LBL PS/Ag. The resulting Raman spectra are shown in Figure 8. The characteristic Raman peaks of DMZ were obviously observed. When the concentration went down to 0.01 ppm, the feature peaks of DMZ were still observed. The UV–Vis absorption measurement method has been well established for DMZ quantification [43]. The UV-vis spectra shown in Appendix A were used to detect DMZ, in which the accurate concentration of DMZ could be determined using the equation established in Appendix A. For detecting DMZ in real samples by UV-Vis spectroscopy, 2 μL of juice from pork was diluted to 2 mL in a quartz cuvette. The pork sample with 100 ppm DMZ was measured by UV-vis spectroscopy (Appendix A) for comparison with the method developed at this research. The concentration determined using the UV-vis spectral method was 95 ppm, and the value determined with the SERS method proposed here is 112 ppm, which means the accuracy of the SERS method is acceptable for detecting DMZ in pork. After washing with water, the feature Raman peaks of DMZ still could be measured from the pork as shown in Appendix A. These results indicates that the LBL PS/Ag substrate could be used as a simple and accurate platform for sensing DMZ in fish by SERS.

## 4. Conclusions 

A controllable synthesis of PS spheres via the EFEP method was successfully achieved by adjusting the polymerization reaction time, ionic strength and concentration of the ionic copolymer. A simple LBL assembly method was employed to fabricate 2D PS/Ag multilayer SERS substrates with a controllable enhancement effect. The 10 layers 2D PS/Ag substrate showed the best SERS enhancement, which was nearly two times stronger compare with the colloidal PS/Ag alone. More importantly, the 2D PS/Ag substrate assembled by LBL showed excellent reproducibility, in which the RSD value was 3.7%. The LBL PS/Ag substrate shows high activity for SERS, that can detect MG and DMZ at low concentrations. Furthermore, the LBL PS/Ag substrate could detect DMZ in real pork samples, and the recovery value was 80%. This 2D SERS substrate using PS spheres and plasmonic NPs can be applied as an effective biosensor for detecting other kinds of pesticides residues in food samples to enhance food quality.

## Figures and Tables

**Figure 1 polymers-13-03270-f001:**
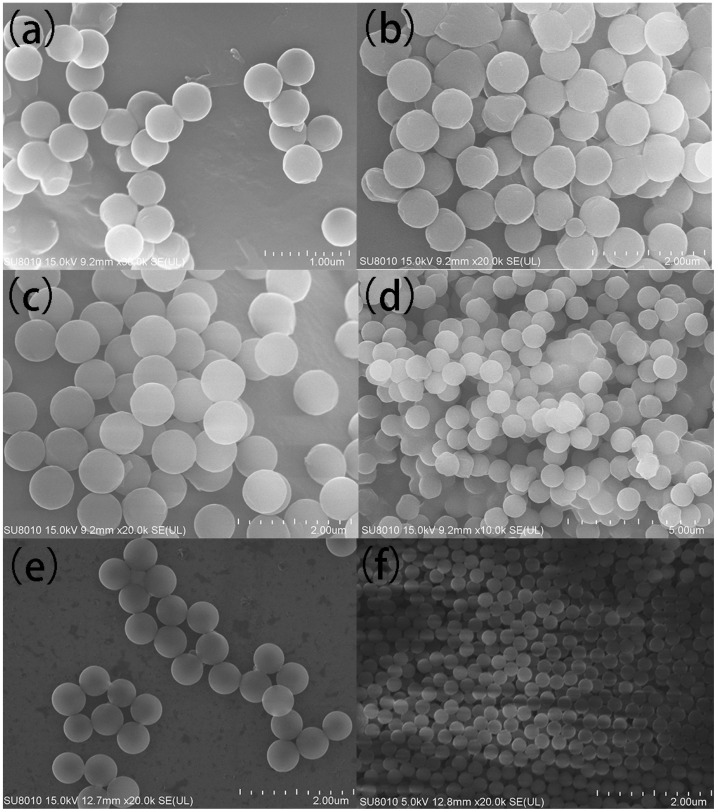
SEM images of polystyrene spheres at different concentration of KHCO_3_ and C_8_H_7_SO_3_Na, ((**a**), 4.5 mM/0.5 mM; (**b**), 7.5 mM/0.5 mM; (**c**), 10 mM/0.5 mM; (**d**), 17 mM/0.5 mM; (**e**), 17 mM/1 mM; (**f**), 17 mM/2 mM).

**Figure 2 polymers-13-03270-f002:**
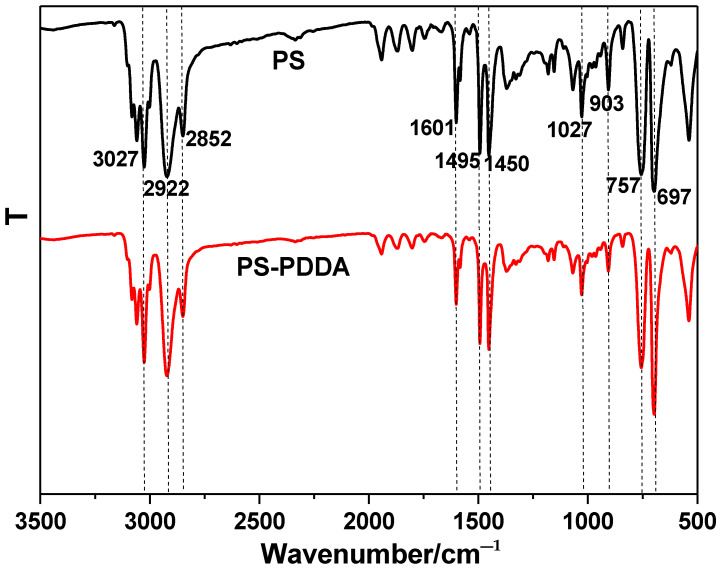
FTIR spectra of the PS spheres before and after PDDA modification.

**Figure 3 polymers-13-03270-f003:**
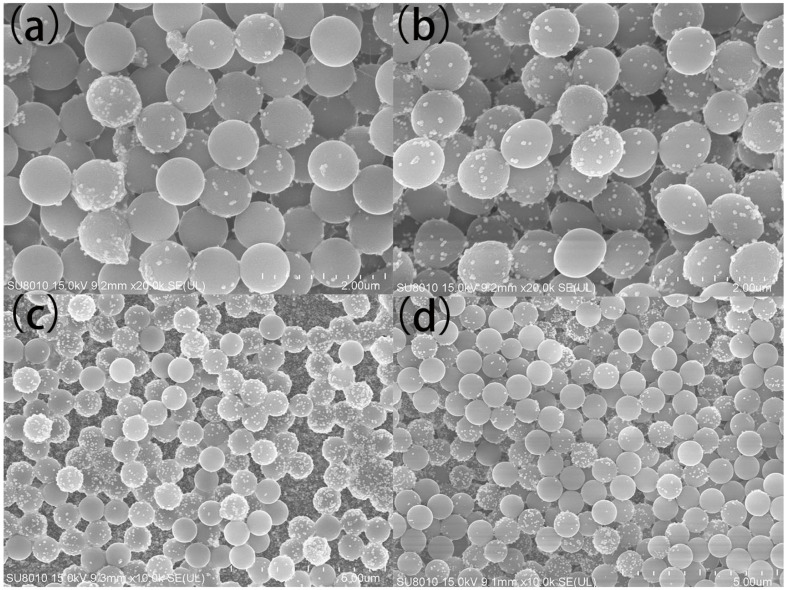
SEM images of 2D PS/Ag substrate assembled by LBL (**a**) 4 layers, (**b**) 6 layers, (**c**) 10 layers and (**d**) 15 layers.

**Figure 4 polymers-13-03270-f004:**
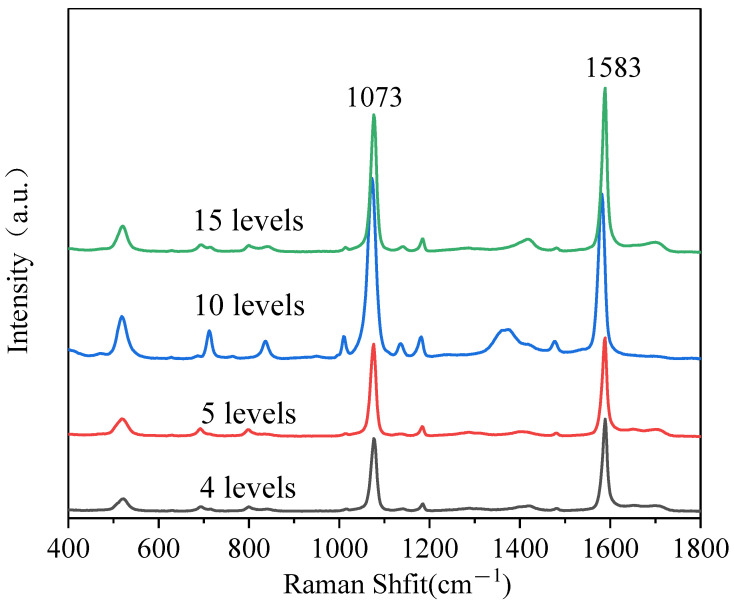
Raman spectra of MBA (100 ppm) on 2D PS/Ag with different layers. Portable Raman spectrometer, laser λ = 785 nm, acquisition time: 2 s.

**Figure 5 polymers-13-03270-f005:**
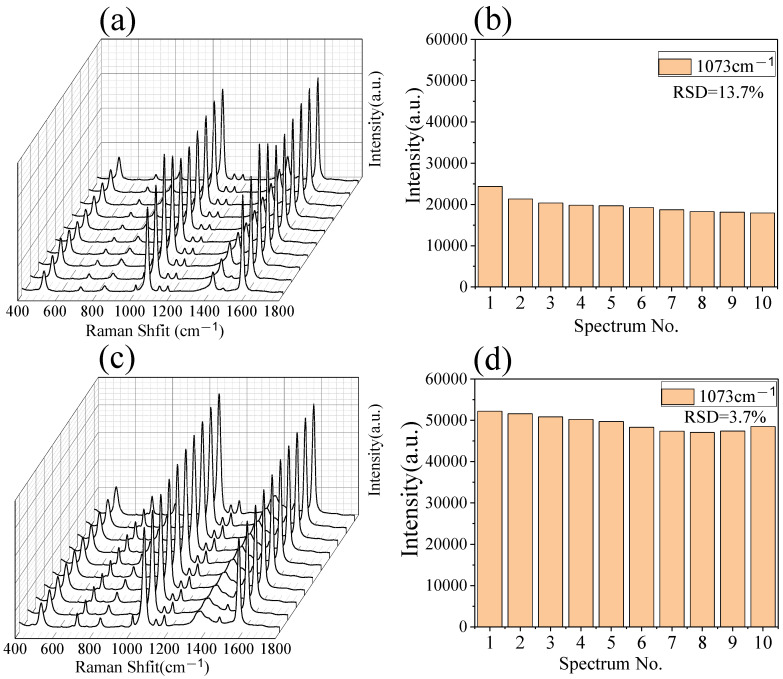
SERS spectra of MBA (100 ppm) measured from 10 random spots from the colloidal PS/Ag (**a**) and 2D PS/Ag assembled by LBL (**c**) substrate, and the corresponding statistical histogram of the intensity of peak at 1073 cm^−1^ (**b**) and (**d**). Portable Raman spectrometer, laser λ = 785 nm, acquisition time: 2 s.

**Figure 6 polymers-13-03270-f006:**
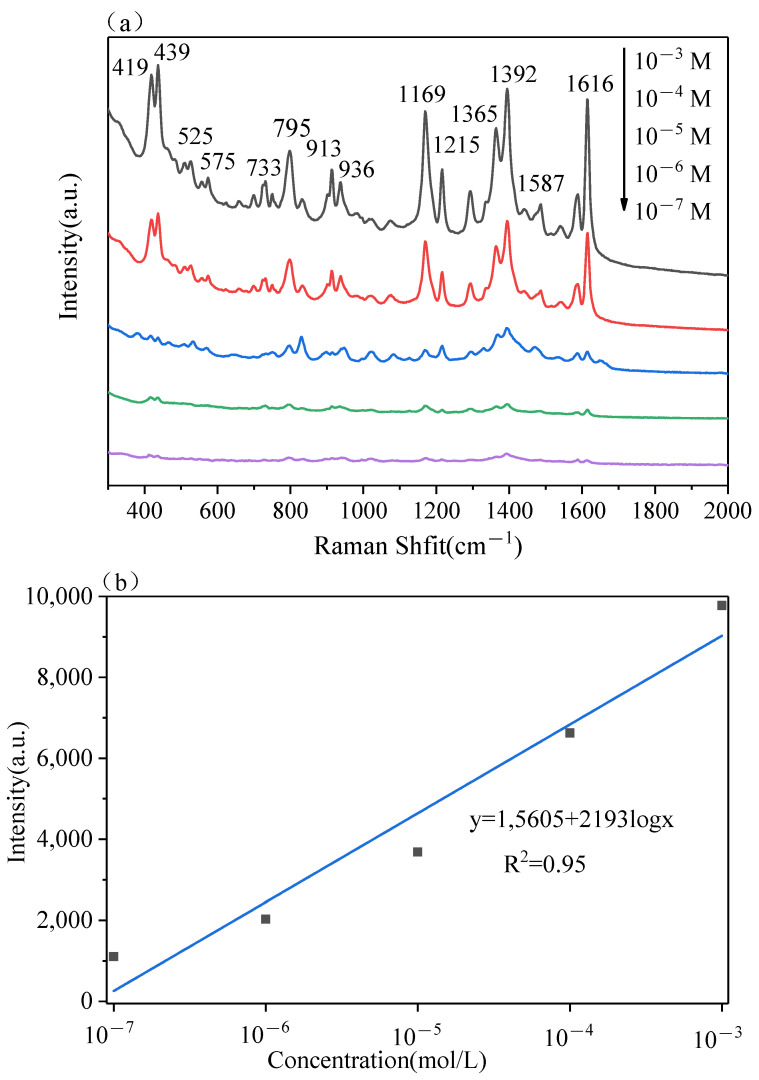
SERS spectra of MG with different concentrations from the LBL PS/Ag substrate (**a**), dose-response curves of the Raman peaks at 796 cm^−1^ (**b**). Portable Raman spectrometer, laser λ = 785 nm, acquisition time: 2 s.

**Figure 7 polymers-13-03270-f007:**
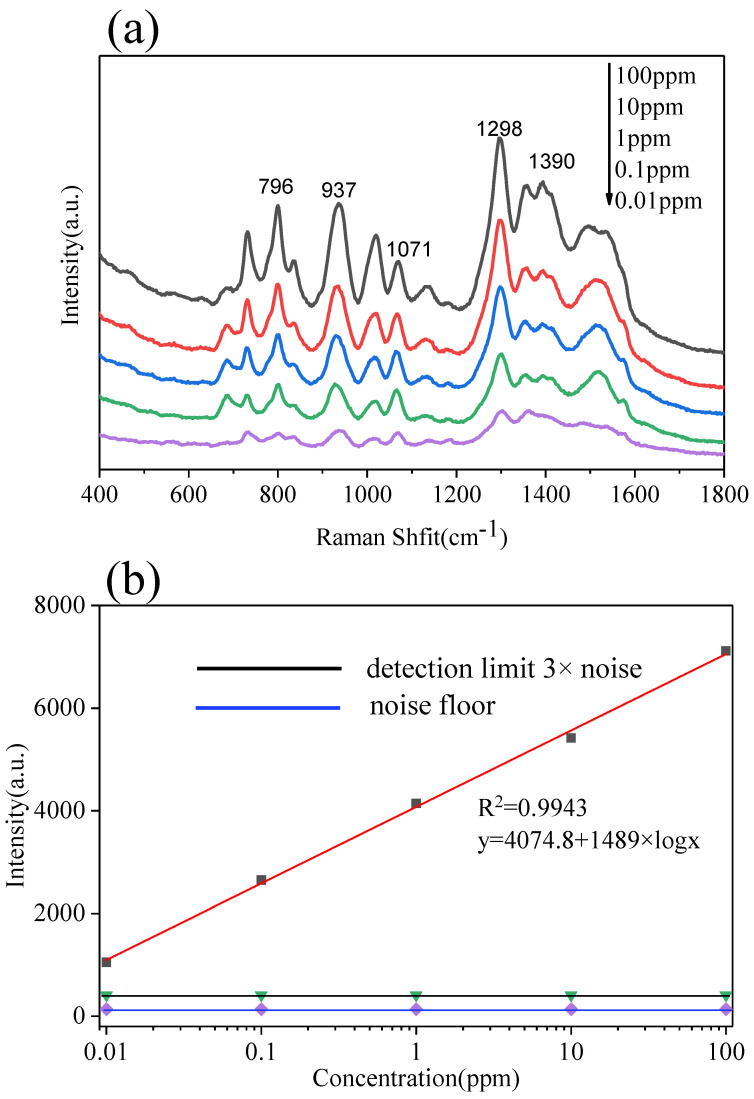
SERS spectra of DMZ at different concentrations from the LBL PS/Ag substrate (**a**), Raman intensity as a function of the logarithm of DMZ concentration (**b**). Portable Raman spectrometer, laser λ = 785 nm, acquisition time: 2 s.

**Figure 8 polymers-13-03270-f008:**
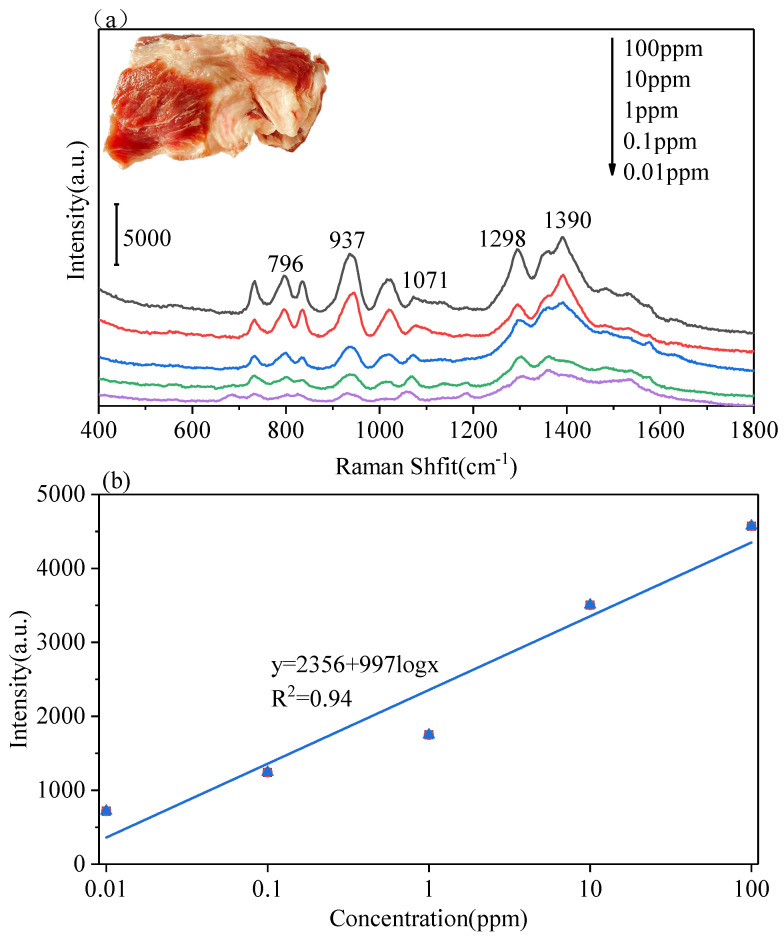
Detection of the DMZ at different concentrations from pork using the LBL PS/Ag substrate(**a**), Raman intensity as a function of logarithm of DMZ concentration from pork (**b**). Portable Raman spectrometer, laser λ = 785 nm, acquisition time: 2 s.

## Data Availability

The data presented in this study are available on request from the corresponding author.

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
