# Peer review of "Layer-by-Layer Assembly of Polystyrene/Ag for a Highly Reproducible SERS Substrate and Its Use for the Detection of Food Contaminants"

_polymers, 2021, doi:10.3390/polym13193270_

Round 1

Reviewer 1 Report

This manuscript reports a combination of Layer-by-Layer technique together with Ag NPs and polyelectrolytes to fabricate a SERS active substrate. The synthesis and dyes SERS part seem to be convincing. I suggest publishing  after revisions noted:

- Literature review is not balanced. First, the authors are suggested to give a proper credit to the Layer-by-layer assembly developed by Yuri Lvov et al. The following paper is suggested for citation:

Hua et al, Langmuir 2002, 18, 17, 6712–6715 

Next, the SERS papers highly relevant to this study are also suggested for citation:

Kahraman et al, 2009, Analytical and bioanalytical chemistry 395 (8), 2559-2567

Kahraman et al, 2010 Analytical chemistry 82 (18), 7596-7602

- the authors are synthesizing the PS spheres, however they are available commercially. Why is it important to synthesize them then?

- what might be environmental implications of yet another case of using nanoplastics?

- the authors need to provide statistical data for results shown in Fig 5 and 6, it is unclear if the results are significantly different

- the pork study case is very preliminary. What is the point in washing pork with the analyte and then measuring? The authors should have found real samples with high analyte content, measure it with another already established technique and then study with their method. Currently, the results in Fig 7 on DMZ detection bear no scientific meaning. This part has to be either performed properly, with all necessary controls and various concentrations of MSZ, confirmed with a different technique. Or removed completely

Author Response

Dear Reviewer

Thank you for reviewing our manuscript (Manuscript ID: polymers-1381713) entitled “Layer-by-layer assembly of polystyrene/Ag for highly reproducible SERS substrate and used for detection of food contaminants.” We appreciate the valuable comments and suggestions from you. We have carefully addressed these concerns and included detailed response in this letter. More experiments were developed and more data were added in this revised manuscript. We believe that the quality of this revised manuscript was improved significantly. The modifications are also highlighted with “Track Changes” function in the resubmitted manuscript.

This manuscript reports a combination of Layer-by-Layer technique together with Ag NPs and polyelectrolytes to fabricate a SERS active substrate. The synthesis and dyes SERS part seem to be convincing. I suggest publishing after revisions noted:

  1. - Literature review is not balanced. First, the authors are suggested to give a proper credit to the Layer-by-layer assembly developed by Yuri Lvov et al. The following paper is suggested for citation:

Hua et al, Langmuir 2002, 18, 17, 6712–6715 

Next, the SERS papers highly relevant to this study are also suggested for citation:

Kahraman et al, 2009, Analytical and bioanalytical chemistry 395 (8), 2559-2567

Kahraman et al, 2010 Analytical chemistry 82 (18), 7596-7602

AnswerThanks for this valuable suggestion, The References (Langmuir 2002, 18, 17, 6712–6715 ; Kahraman et al, 2009, Analytical and bioanalytical chemistry 395 (8), 2559-2567; Kahraman et al, 2010 Analytical chemistry 82 (18), 7596-7602) were cited in the manuscript and more discussion were added.

The films fabricated by LBL technology mainly depend on the electrostatic adsorption between the opposite charges, in which the dimensional precision could down to the submicron level [27]. Kahraman et al. deposited plasmonic NPs on bacterial cells by layer-by-layer method, in which the Raman information of single bacterial cell were successfully obtained [28].

  1. the authors are synthesizing the PS spheres, however they are available commercially. Why is it important to synthesize them then?

AnswerThanks for this valuable comment, there are two reasons for synthesis PS spheres by ourself:

First: we want provide more useful details for the preparation of PS spheres.

Second: The cost of the PS spheres synthesized in lab is much lower than that from the commercial products.

  1. what might be environmental implications of yet another case of using nanoplastics?

AnswerThanks for this valuable comment, the amount of PS spheres used for constructing SERS substrate is small, and after SERS measurement the substrates were burned in 500 ° C to exclude the nanoplastics.

  1. the authors need to provide statistical data for results shown in Fig 5 and 6, it is unclear if the results are significantly different

AnswerThanks for this valuable suggestion. We have replotted the Fig5 b in same scale with Fig 5d, that could show obvious difference.

The statistical data was added in Fig 6. 

  1. the pork study case is very preliminary. What is the point in washing pork with the analyte and then measuring? The authors should have found real samples with high analyte content, measure it with another already established technique and then study with their method. Currently, the results in Fig 7 on DMZ detection bear no scientific meaning. This part has to be either performed properly, with all necessary controls and various concentrations of MSZ, confirmed with a different technique. Or removed completely

AnswerThanks for this valuable suggestion. The washing pork with the DMZ was measured and the data added in Figure S7.

The established technique (UV-vis spectra) was used to detect DMZ and the data was added in Figure S4.

The SERS spectra of various concentrations of DMZ was added in Figure7.

“The LBL PS/Ag was used as SERS substrate for rapidly detecting DMZ from pork.  The solutions of DMZ at different concentrations were sprayed onto the surface of pork.  The 5 μL of juice from pork was spotted onto the LBL PS/Ag, the Raman spectra were shown in Figure 8. The characteristic Raman peaks of DMZ were observed obviously. When the concentration down to 0.01 ppm, the feature peaks of DMZ still observed. The UV-vis spectra as shown in Figure S4 were used to detect DMZ, in which the accurate concentration of DMZ could be determined as the equation established in Figure S4b. For detecting DMZ from real sample by UV-vis spectra, 2 μL of juice from pork was diluted to 2 mL in quartz cuvette. The pork sample with 100 ppm DMZ was measured with UV-vis spectra (Figure S5) for comparison with the method developed at this research. The concentration determined from UV-vis spectral method was 95 ppm, and the value determined from the SERS method proposed at here is 112 ppm, which mean the accuracy of the SERS method is acceptable for detecting DMZ from pork. After washing with water, the feature Raman peaks of DMZ still could be measured form the pork as shown in Figure S6. These results indicates that the LBL PS/Ag substrate could be used as a simple and accurate platform for sensing DMZ from fish by SERS.”

 Sincerely

Xianming Kong

Associate Professor

School of Petrochemical Engineering

Liaoning Petrochemical University,

Fushun, Liaoning 113001, P. R. China,

E-mail: xmkong@lnpu.edu.cn

Reviewer 2 Report

The work " Layer-by-layer assembly of polystyrene/Ag for highly repro-ducible SERS substrate and used for detection of food contami-nants " presents a representative analysis of the proposed study, and I recommend publication after the following corrections:

The sections "introduction, methodology and results and discussion" should be better divided. The introduction should include the justifications for the use of pork (and not the results) and also the problem of DMZ for human health; The methodology must contain the conditions of all the experiments carried out, and if the authors deem it necessary, this information can be retrieved in the discussion of the results; The results discussion section lacks a literature review, reducing the credibility of the analyzes and, consequently, of the manuscript.

Minor changes should also be considered:

1) In the summary, define the abbreviation "SERS"

2) In section 2.3 the unit "ml" must be "mL". The unit must be separated from the value (72 ° C) there are other occurrences in the other sections, check the entire document.

3) Standardize the unit of time throughout the document, express "h or hour"

4) In section 3.2 the unit “uL” must be “μL”.

Author Response

Dear Reviewer

Thank you for reviewing our manuscript (Manuscript ID: polymers-1381713) entitled “Layer-by-layer assembly of polystyrene/Ag for highly reproducible SERS substrate and used for detection of food contaminants.” We appreciate the valuable comments and suggestions from you. We have carefully addressed these concerns and included detailed response in this letter. More experiments were developed and more data were added in this revised manuscript. We believe that the quality of this revised manuscript was improved significantly. The modifications are also highlighted with “Track Changes” function in the resubmitted manuscript.

the work " Layer-by-layer assembly of polystyrene/Ag for highly repro-ducible SERS substrate and used for detection of food contami-nants " presents a representative analysis of the proposed study, and I recommend publication after the following corrections:

The sections "introduction, methodology and results and discussion" should be better divided. The introduction should include the justifications for the use of pork (and not the results) and also the problem of DMZ for human health; The methodology must contain the conditions of all the experiments carried out, and if the authors deem it necessary, this information can be retrieved in the discussion of the results; The results discussion section lacks a literature review, reducing the credibility of the analyzes and, consequently, of the manuscript.

AnswerThanks for this valuable suggestion, the sentence about justifications for the use of pork and the problem of DMZ for human health was move to the introduction from results.

More literatures were added in the part of results (ref: 32, 33, 34, 35, 38, 39, 42 and 43).

The conditions of all the experiments were added in the manuscript

Such as:

Fourier transform infrared (FTIR) spectra were collected on a Nicolet 6700 spectrometer (PerkinElmer, USA) with DTGS detector and KBr method, typically 32 scans were collected to obtain a satisfactory signal-to-noise ratio with the resolution 4 cm-1.

5 μL of MBA solution was dropped onto the surface of LBL PS/Ag substrate.

10 μL of colloidal PS/Ag was dropped onto glass substrate and dried in air condition, 5 μL of MBA solution (100 ppm) was dropped onto the colloidal PS/Ag and used for Raman measurement.

Minor changes should also be considered:

1) In the summary, define the abbreviation "SERS"

AnswerThanks for this valuable suggestion, the full name of SERS was defined in the first pressence in abstract.

2) In section 2.3 the unit "ml" must be "mL". The unit must be separated from the value (72 ° C) there are other occurrences in the other sections, check the entire document.

AnswerThanks for this valuable suggestion, we have checked all through the manuscript and corrected the units.

3) Standardize the unit of time throughout the document, express "h or hour"

AnswerThanks for this valuable suggestion, all unit of time was corrected to ‘h’

4) In section 3.2 the unit “uL” must be “μL”.

AnswerThanks for this valuable suggestion, the unit “uL” was corrected to “μL”

Sincerely

Xianming Kong

Associate Professor

School of Petrochemical Engineering

Liaoning Petrochemical University,

Fushun, Liaoning 113001, P. R. China,

E-mail: xmkong@lnpu.edu.cn

Round 2

Reviewer 1 Report

The revised MS can be accepted